# Monitoring, surveillance, antimicrobial resistance and genetic diversity analysis of non-typhoidal *Salmonella* in South Africa from 1960–2023 from animal and animal products

Itumeleng Matle[1,2]*, Davies Mubika Pfukenyi[3], Nozipho Maphori[1,4]
Nkagiseng Moatshe[1,2] Thabo Nkabinde[1,5], Annah Motaung[1,2], Tracy Schmidt[6],
Emmanuel Seakamela[1], Mulunda Mwanza[4], Lubanza Ngoma[4], Mohamed Sirdar[7,8],
Khanyisile R. Mbatha[9], Kudakwashe Magwedere[10]

**1** Bacteriology Division, Agricultural Research Council, Onderstepoort Veterinary Research, Onderstepoort, South Africa, **2** Department of Agriculture and Animal Health, University of South Africa, Science Campus, Florida, South Africa, **3** Faculty of Animal and Veterinary Sciences, Botswana University of Agriculture and Natural Resources, Gaborone, Botswana, **4** Department of Animal Health, Faculty of Natural and Agriculture, Sciences, Northwest University, Mafikeng, South Africa, **5** Tshwane University Technology Main campus, Pretoria West, Pretoria, South Africa, **6** Allerton Provincial Veterinary Laboratory, KwaZulu-Natal Department of Agriculture and Rural Development, Pietermaritzburg, South Africa, **7** International Society for Infectious Diseases (ISID), Boston, United States of America, **8** Faculty of Veterinary Science, University of Pretoria, Onderstepoort, South Africa, **9** School of Interdisciplinary Research and Graduate Studies, College of Graduate Studies, University of South Africa, Pretoria, South Africa, **10** Boston University, Center on Emerging Infectious Diseases, Boston, Massachusetts, United States of America

* Itumelengmatle@gmail.com

## Abstract

Salmonellosis remains one of the most frequently reported foodborne diseases globally, with the highest burden in low-resource areas. The millions of deaths caused by Nontyphoidal *Salmonella* (NTS) infections emphasize the urgent need for timely, detailed, and evidence-based interventions to effectively manage and monitor NTS burdens. This study retrospectively analyzed 1,028 NTS isolates from animals, the environment, and food products in South Africa, collected between 1960 and 2023. Among the 102 serotypes identified, *S*. Heidelberg, isolated only between 2000−2009 and 2020−2023, accounted for 94.3% of isolations during the latter period, suggesting a recent shift in *Salmonella* epidemiology in the region. The highest resistance rates were observed for cefoxitin (65.7%), cephalothin (62.8%), and tetracycline (59.8%), with a significant increase in resistance to several antibiotics, including ceftriaxone and aztreonam, from 2010−2023. Genetic analysis revealed that *S*. Gallinarium had the highest prevalence of antibiotic resistance genes, such as *tetA* (71.4%), *qnrA* (64.3%), *cat*1 (64.3%), $bla_{PSE}$ (57.1%), and both $bla_{CMY-2}$ and *qnrB* at 50%. The $bla_{PSE}$ and $bla_{SHV}$ genes were strongly associated with ceftriaxone resistance in *S*. Dublin isolates, while $bla_{PSE}$ and *qnrS* were linked to chloramphenicol resistance in *S*. Enteritidis and *S*. Dublin isolates. Additionally, 87% of the virulence genes screened were present in over 50% of the serotypes, indicating increased adaptability and

**Data availability statement:** This is not applicable to our research, as the metadata used in this study were obtained from laboratory records and compiled in Microsoft Excel. Metadata can be included as part of the supplementary Data are available from the Dr Lia Rotherhan Ethics Committee (contact via RotherhanL@arc.agric.za or +2712 5299111) for researchers who meet the criteria for access to confidential data.

**Funding:** This study was partially funded by the Department of Agriculture, Land Reform and Rural Development (DALRRD), and partially North-West University. The funders had no role in study design, data collection and analysis, decision to publish, or preparation of the manuscript.

**Competing interests:** The authors have declared that no competing interests exist.

potential shifts in disease dynamics. The rise in antimicrobial resistance, driven by antimicrobial misuse, horizontal gene transfer, and biofilm formation, could alter serotype dynamics and changing disease epidemiology. This trend underscores the urgent need for effective antimicrobial stewardship and surveillance to combat the spread of antibiotic resistance in *Salmonella* populations.

## Introduction

*Salmonella* is a zoonotic bacterial pathogen that poses significant risks to both human and animal health [1]. Currently, around 2700 different *Salmonella* serotypes have been identified [2] and many of which can cause cross-infection between animals and humans [3]. Ninety nine percent of human serotypes belong to *Salmonella enterica* subsp. *enterica,* which is associated with causing invasive Nontyphoidal *Salmonella* (iNTS) infections. Most cases of iNTS disease are documented in sub-Saharan Africa, where yearly incidence rates range from 175 to 388 cases per 100,000 person-years, and case fatality rates as high as 25% have been reported in young children [4]. Various animals, especially those intended to be consumed as part of the human diet, have been identified as reservoirs for iNTS serotypes [5].

Several studies have reported a concerning rise in antibiotic resistance among *Salmonella* strains, particularly against first-line antimicrobials, with multidrug-resistant (MDR) variants contributing to increased morbidity and mortality [6–8]. The emergence of extended-spectrum β-lactamase (ESBL)-producing and fluoroquinolone-resistant *Salmonella* strains has significantly undermined the effectiveness of available treatments. Among the ESBL families detected in clinical settings, $bla_{TEM}$, $bla_{SHV}$, and especially $bla_{CTX-M}$ are the most commonly identified in *Salmonella* species [9,10]. In addition to antimicrobial resistance, the pathogenicity of *Salmonella* is driven by a complex interplay of chromosomal and plasmid-encoded virulence factors that collectively enable the bacterium to effectively colonize the host, evade immune responses, and cause disease. These virulence factors are essential for key processes such as adhesion to epithelial cells, invasion of host tissues, intracellular replication, and eventual dissemination, all of which contribute to tissue damage and clinical symptoms [11,12].

Active and passive monitoring and surveillance systems have been operational in South Africa (SA) prior to 1958, initially relying on data from human medical laboratories [13]. The *Salmonella* Typing Unit of the South African Institute for Medical Research (SAIMR) established in 1952 had identified 132 serotypes by 1958. Today, structured passive and active public health surveillance continues under the coordination of the National Institute for Communicable Diseases (NICD). The notifiable disease surveillance system initiated in the 1970s and, later amended, facilitates the reporting of *Salmonella* cases through the notifiable medical conditions surveillance system, enabling the audit and follow-up of isolates and their associated metadata [14].

Onderstepoort Veterinary Research (formerly the Onderstepoort Veterinary Institute) has been conducting *Salmonella* serotyping in SA since the 1930s from

animal and food sources [15,16]. Apart from a few reports, the literature contains the description of a number of *Salmonella* types originally isolated from animals and animal products In South Africa, the Animal Diseases and Parasites Act (1956), repealed by the Animal Diseases Act, (Act No 35 of 1984) mandated that *S.* Enteritidis, *S.* Pullorum and *S.* Gallinarum isolated from all fowls, including poultry and ostriches, must be reported to the state authority and the latter two serotypes were identified as notifiable diseases. Extensive monitoring and surveillance of *Salmonella* serotypes has been ongoing under the Meat Safety Act (MSA), 2000 (Act No. 40 of 2000) which provides measures to promote meat safety and the safety of animal products. The largest official data are generated from monitoring and verification of imported meat at the ports of entry. Although, there has been no nationwide objective and systematic evaluation of the monitoring and surveillance of NTS in SA since inception of the fragmented surveillance programs; in 2023, a formal integrated national monitoring and surveillance of foodborne pathogens inclusive of NTS was established under the Meat Safety Act 2002.

Given the escalating incidence and recurrence of *Salmonella* outbreaks in both human and animal populations, alongside heightened awareness of the imperative to address the burgeoning challenge of antibiotic resistance in *Salmonella*, there is a pressing need for the implementation of longitudinal monitoring and surveillance programs. Such initiatives are essential for providing important epidemiological data which is critical for risk assessment and disease control strategies, identify trends, reservoirs and transmission pathways of various *Salmonella* serotypes. The purpose of this communication is to record the antimicrobial resistance and genetic diversity among *Salmonella* serotypes isolated in SA from 1960 to 2023 which has become a matter of considerable public health and safe trade concern.

## Materials and methods

### Study design and source of isolates

The current retrospective cohort study was conducted using government and private laboratory-confirmed *Salmonella* isolates that were stored at the Agricultural Research Council: Onderstepoort Veterinary Research (ARC: OVR)-General Bacteriology Laboratory for a period of 63 years (1960–2023). As of 2018, the Veterinary Procedural Notification 56 (https:www.dalrrd.gov.za/vpn56/2019) mandates that all *Salmonella* strains isolated from meat samples in SA be forwarded to the ARC: OVR-General Bacteriology Laboratory for storage and molecular typing monitoring. Consequently, isolates utilized in this study were recovered from samples collected from various provinces of SA including ports of entry and from diverse sources such as environments, feed, and animal products.

### Revival and confirmation of *Salmonella* isolates

*Salmonella* isolates were either lyophilized or preserved in glycerol. These isolates had previously been identified and serotyped at the ARC: OVR Bacteriology Laboratory using the White-Kauffmann-Le Minor Scheme [2]. To revive the isolates, they were inoculated into 2 ml of Brain Heart Infusion broth (Oxoid, Thermofisher Scientific, Johannesburg, SA) and then incubated at 37 ℃ for 18–24 hours. Subsequently, the BHI broth was subcultured onto nutrient agar (Oxoid, Thermofisher Scientific, Johannesburg, SA) and further incubated at 37 ℃ for 18–24 hours. Confirmation of *Salmonella* isolates was performed using polymerase chain reaction (PCR). The DNA template for PCR was extracted using a slightly modified boiling method, as described by [17].

### PCR confirmation of *Salmonella* isolates

*Salmonella* isolates underwent confirmation through the amplification of the *invA* (224 bp) gene (F:5'-ACAGTGCTCG TTTACGACCTGAAT-3; R:5'-AGACGACTGGTACTGATCTAT-3') , known for its genus specificity, and the *iroB* (606 bp) gene, (F: 5'-TGCGTATTCTGTTTGTCGGTCC-3; R: 5'-TACGTTCCCACCATTCTTCCC-3) as previously described by [18].

## Antimicrobial susceptibility testing

All confirmed *Salmonella* isolates underwent antimicrobial susceptibility testing (AST) using the Kirby Bauer disk diffusion method [19]. The antimicrobial panels included the following antibiotics and disk content: cefoxitin 30 μg (FOX), cephalothin 30 μg (KF), tetracycline 30 μg (TET), piperacillin-tazobactam 40 μg (TZB), ciprofloxacin 5 μg (CIP), amoxicillin-clavulanic acid 30 μg (AMC), ceftriaxone 30 μg (CRO), amikacin 30 μg (AK), gentamicin 10 μg (GEN), ampicillin 10 μg (AMP), sulfamethoxazole-trimethoprim 25 μg (STX), chloramphenicol 30 μg (CHL), aztreonam 30 μg (ATM) (Thermo-Fisher Scientific, USA). The overnight pure cultures of *Salmonella* on nutrient agar were inoculated into sterile saline. The bacterial suspension was inoculated aseptically onto Mueller-Hinton agar plates (Thermo-Fisher Scientific, USA) following which antibiotic discs were placed on the surface. Inverted plates were inverted and incubated at 37 °C for 24 hrs whereafter zone diameters were measured and recorded.

## Molecular detection of antibiotics resistance-associated genes

Multiple multiplex PCR assays were conducted to detect the presence of 18 resistance genes, encompassing beta-lactamase resistance ($bla_{TEM}$, $bla_{CMY-2}$, $bla_{SHV}$, and $bla_{PSE}$); sulphonamide resistance (*sul1, sul2,* and *sul3*); phenicol resistance (*cat1, flo,* and *cmlA*); trimethoprim resistance (*dfrI, dfrXII*, and *dfrXIII*); tetracycline resistance (*tetA* and *tetB*) and quinolone resistance (*qnrA, qnrB,* and *qnrS*). Each PCR reaction mixture (25 μL) consisted of 12.5 μL of Taq 2x Master Mix RED (Ampliquor, Denmark), 2 μL (5 pmol/ μL) of each primer (Inqaba Biotechnical Industries (Pty) Ltd., SA), 4.5 μL of UltraPure DNase/RNase-Free Distilled Water (ThermoFisher Scientific, USA), and 4 μL of DNA template. Amplification was carried out using a GeneAmp 9700 PCR machine (Applied Biosystems, USA) under the following conditions: initial denaturation at 94 °C for 3 min, followed by 30 cycles of denaturation at 94 °C for 30 seconds, annealing at variable temperatures (S1 Table) for 30 seconds, extension at 72 °C for 1 min, and final extension at 72 °C for 10 min. The PCR products were then subjected to electrophoresis on a 3% agarose gel containing 4 μL of ethidium bromide. Gel visualization was performed using UV light, and images were captured using a gel documentation system (OmegaFlour, Vacutec).

## Determination of virulence genes

Screening for virulence genes was also performed on all *Salmonella* isolates. The following virulence genes were tested: *invA, sopB, gtgB, sspH1, sopE, spvC, pefA, sifA, gipA, sodC1, gtgE, mig5, sspH2, rck,* and *srgA*. A slightly altered procedure previously described by Capuano et al. (2013) was used. The 25ul PCR reaction mixture comprised 12.5 μL of Taq 2x Master Mix RED (Ampliquor, Denmark), 2 μL (5 pmol/ μL) of each primer (Inqaba Biotechnical Industries (Pty) Ltd., SA), 4.5 μL UltraPure DNase/RNase-Free Distilled Water (ThermoFisher Scientific, USA) and 4 μL DNA template. GeneAmp 9700 (Applied biosystems, USA) PCR machine was used for amplification using the following PCR conditions: initial denaturation at 94 °C for 3 min, 30 cycles of denaturation at 94 °C for 30 seconds, annealing temperature variable (S2 Table) for 1 min, extension at 72 °C for 1 min, and final extension at 72 °C for 5 min. PCR amplicons were analysed by electrophoresis on a 1.5% agarose gel containing 4 μL ethidium bromide, the gel was observed using UV light and photographed (OmegaFlour, Vacutec).

## Data analysis

The data recording, editing, and generation of descriptive statistics were conducted using Microsoft Excel®. The prevalence of each serotype was calculated as a percentage of its contribution to the total number of *Salmonella* isolates during the study period, along with its corresponding 95% confidence interval. The prevalence of antibiotic resistance genes identified among the *Salmonella* serotypes and isolates was determined as the ratio of the number of genes to the total number of serotypes or isolates, expressed as a percentage. Similarly, the prevalence of virulence genes was calculated based on the total number of serotypes or isolates and expressed as a percentage. For serotypes, antibiotic resistance

genes, and virulence genes, prevalence calculations were restricted to those with ≥ 16 isolates. Antibiotic resistance for each antibiotic tested (n = 13) was determined as the ratio of isolates resistant to the antibiotic to the total number of isolates, expressed as a percentage. The overall antibiotic resistance of serotypes and isolates, as well as the antibiotic resistance of isolates for serotypes with ≥ 10 isolates, were calculated.

To investigate temporal patterns of *Salmonella* serotypes over the 63-year period, the years were grouped into 6 different periods consisting of 10-year intervals: (1960–1969), (1970–1979), (1980–1989), (1990–1999), (2000–2009) and (2010–2019). Note that the data before 1960 and after 2020 are not representative of a full decade. Temporal patterns were examined for *Salmonella* serotypes with ≥ 16 isolates, and the number of isolates and their percentage contribution for each 10-years period were calculated. Regardless of *Salmonella* serotype, the percentage of antibiotic resistance for each antibiotic was calculated for each 10-years period. Comparisons were made across the 10-year period categories, utilizing the statistical package EpiCalc 2000 (version 2), to measure percentage differences between study periods. The p values (analysed by the Chi-Square test for proportions) less than 0.05 were considered significant.

The association between antibiotic resistance and the presence of antibiotic resistance genes was assessed by calculating the relative risk (RR) using the EpiCalc 2000 (version 2) statistical package. Additionally, comparisons between antibiotic resistance and the animal origin of isolates were conducted using the statistical tool EpiCalc 2000 (version 2) to measure percentage differences, with p values < 0.05 considered significant.

### Ethics statement

The protocol for this study was approved by the Ethics Committee of the Onderstepoort Veterinary Research and University of South Africa prior to the start of the study (approval number: 2022/CAES_AREC/059.

## Results

### Prevalence of *Salmonella* serotypes

During the study period, a total of 102 *Salmonella* serotypes were identified from 1028 isolates. *Salmonella* Heidelberg was the most prevalent serotype, accounting for (22.2%; n = 228) of the isolates, followed by *S*. Typhimurium (21.0%; n = 216), *S*. Enteritidis (13.4%; n = 138), and *S*. Minnesota (10.5%; n = 108) (Table 1). Sixteen (1.6%) of the isolates could not be serotyped as they reacted to the anti-*Salmonella* polyvalent somatic antisera. The majority (77.5%; 797/1028) of the isolates belonged to only 9% (9/102) of the serotypes while 91.2% of serotypes (n = 93) exhibited a prevalence of less than 1%, with more than half (71%, 66/93) of them consisting of only two or a single isolate (Table 1).

### Source of the *Salmonella* isolates

During the period under review, a total of 49.9% (n = 513) isolates were recovered from animals, 32.1% (n = 330) from meat, 12.9% (n = 133) from non-meat food sources, 2.7% (n = 28) from feed, 1.5% (n = 15) from environmental samples, and 0.9% (n = 9) from unknown sources. Despite the involvement of a large number of serotypes (102), the distribution of the top nine serotypes indicated that the majority of isolates originated from animal sources (n = 385), with *S*. Typhimurium being the most prevalent (41.8%; n = 161), followed by *S*. Enteritidis (25.5%; n = 98) (Table 2). A similar pattern was observed in feed samples, where *S*. Typhimurium was the most frequently detected serotype, accounting for 61.5% (n = 16) of the isolates, followed by *S*. Enteritidis at 7 (26.9%). However, a contrasting trend was observed in isolates from non-meat food sources, where *S*. Enteritidis was the most detected serotype, with 53.2% (n = 25) of the isolates, followed by *S*. Typhimurium at 31.9% (n = 15). In meat samples, the predominant serotypes were *S*. Heidelberg and *S*. Minnesota, with *S*. Heidelberg being the most frequently detected at 61.4% (n = 188) followed by *S*. Minnesota at 27.8% (n = 85) (Table 2).

**Table 1. Prevalence of *Salmonella* serotypes.**

| Serotype | Number isolated (n = 1028) | Prevalence (95% CI) |
|---|---|---|
| *S.* Heidelberg | 228 | 22.2 (19.7-24.9) |
| *S.* Typhimurium | 216 | 21.0 (18.6-23.7) |
| *S.* Enteritidis | 138 | 13.4 (11.4-15.7) |
| *S.* Minnesota | 108 | 10.5 (8.7-12.6) |
| *S.* Dublin | 43 | 4.2 (3.1-5.6) |
| *S.* Bovismorbificans | 18 | 1.8 (1.1-2.8) |
| *S.* Infantis | 16 | 1.6 (0.9-2.6) |
| *S.* Gallinarum | 14 | 1.4 (0.8-2.3) |
| *Other serotypes (< 1% prevalence, n = 93) | 231 | 22.5 (20.0-25.2) |

* These include *S.* Hadar (n=9), *S.* Kentucky (n=9), *S.* Vejle (n=9), *S.* Anatum (n=8), *S.* Brancester (n=8), *S.* Orion (n=8), *S.* Virchow (n=8), *S.* Muenchen (n=7), *S.* Saintpaul (n=6), *S.* Thompson (n=6), *S.* Abony (n=5), *S.* Mbandaka (n=5), *S.* Muenster (n=5), *S.* Pretoria (n=5), *S.* Senftenberg (n=5), *Salmonella* II (n=5), *S.* Choleraesuis (n=4), *S.* Gloucester (n=4), *S.* Hull (n=4), *S.* Schwarzengrund (n=4), *S.* Stanley (n=4), *S.* Tennessee (n=4), *S.* Aarhus (n=3), *S.* Aberdeen (n=3), *S.* Ablogane (n=3), *S.* Chester (n=3), *S.* Molade (n=3), *S.* Agona (n=2), *S.* Bardo (n=2), *S.* Bareilly (n=2), *S.* Carran (n=2), *S.* Cleveland (n=2), *S.* Coela (n=2), *S.* Cremieu (n=2), *S.* Derby (n=2), *S.* Lagos (n=2), *S.* London (n=2), *S.* Newport (n=2), *S.* Nigeria (n=2), *S.* Ohio (n=2), *S.* Oritameum (n=2), *S.* Othmarschen (n=2), *S.* Paratyphi (n=2), *S.* Typhi (n=2), *S.* Yaba (n=2), *S.* Abaetetuba (n=1), *S.* Amager (n=1), *S.* Amersfoot (n=1), *S.* Be (n=1), *S.* Blockley (n=1), *S.* Bonn (n=1), *S,* Breden (n=1), *S.* Bron (n=1), *S.* Cardoner (n=1), *S.* Carvalis (n=1), *S.* Colorado (n=1), *S.* Doncaster (n=1), *S.* Durban (n=1), *S.* Dusseldorf (n=1), *S.* Eastbourne (n=1), *S.* Fischerkietz (n=1), *S.* Give (n=1), *S.* Glostrup (n=1), *S.* Haifa (n=1), *S.* Hayindongo (n=1), *S.* Hilversum (n=1), *S.* Isangi (n=1), *S.* Ituri (n=1), *S.* Ivory (n=1), *S.* Javiana (n=1), *S.* Jerusalem (n=1), *S.* Kangi (n=1), *S.* Kottbus (n=1), *S.* Obeden (n=1), *S,* Panama (n=1), *S.* Pullorum (n=1), *S.* Quantum (n=1), *S.* Reading (n=1), *S.* Rissen (n=1), *S.* Sandiego (n=1), *S.* Stockholm (n=1). *S.* Soahanina (n=1), *S.* Taksony (n=1), *S.* Tambacounda (n=1), *S.* Tees (n=1), *S.* Tibati (n=1), *S.* Virgina (n=1), *S.* Wangata (n=1), *S.* Westhampton (n=1), *S.* Wippra (n=1), *S.* Yoruba (n=1), *S.* Zanzibar (n=1).

**Table 2. Source of isolates and distribution of the major serotypes 1960–2023.**

| Isolation Source | Isolates Tested | Number of isolates tested for major serotypes | Serotypes (%) | | | | | | | |
|---|---|---|---|---|---|---|---|---|---|---|
| | | | *S.* Bovis-Morbificans | *S.* Dublin | *S.* Enteritidis | *S.* Gallinarum | *S.* Heidelberg | *S.* Infantis | *S.* Minnesota | *S.* Typhimurium |
| Animal | 513 | 385 | 16 (4.2) | 43 (11.2) | 98 (25.5) | 13 (3.4) | 32 (8.3) | 3 (0.8) | 19 (4.9) | 161 (41.8) |
| Environment | 15 | 12 | 0 (0.0) | 0 (0.0) | 1 (8.3) | 0 (0.0) | 1 (8.3%) | 0 (0.0) | 1 (8.3) | 9 (75.0) |
| Feed | 28 | 26 | 0 (0.0) | 0 (0.0) | 7 (26.9) | 0 (0.0) | 3 (11.5) | 0 (0.0) | 0 (0.0) | 16 (61.5) |
| Meat | 330 | 306 | 0 (0.0) | 0 (0.0) | 5 (1.6) | 0 (0.0) | 188 (61.4) | 13 (4.2) | 85 (27.8) | 15 (4.9) |
| Non-meat food source | 133 | 47 | 2 (4.3) | 0 (0.0) | 25 (53.2) | 1 (2.1) | 2 (4.3) | 0 (0.0) | 2 (4.3) | 15 (31.9) |
| Unknown | 9 | 4 | 0 (0.0) | 0 (0.0) | 3 (75.0) | 0 (0.0) | 0 (0.0) | 0 (0.0) | 1 (25.0) | 0 (0.0) |
| **Total** | **1028** | **780** | **18 (2.3)** | **43 (5.5)** | **139 (17.8)** | **14 (1.8)** | **226 (29.0)** | **16 (2.1)** | **108 (13.8)** | **216 (27.7)** |

## Temporal patterns of the *Salmonella* isolates

Fig 1 depict the temporal distribution of both the total isolates (irrespective of serotype) and serotypes with ≥ 16 isolates isolated during the study period. *S.* Typhimurium was consistently isolated across all period intervals, demonstrating a gradual temporal increase over the years, with 70.4% being isolated between 2000–2009, and 2010–2019 (Fig 1a). *S.* Heidelberg isolates were isolated starting from the 2000–2009 period with 93.4% of them isolated during the 4-year 2020–2023 period (data not shown). No isolates of *S.* Enteritidis were detected between 1960–1969 and it showed a notable peak between 1990–1999. *S.* Minnesota isolates started to be isolated during the 2000–2009 period (Fig 1a) with a significant surge during the 4-year 2020–2023 period (data not shown) where 94.4% (102/108) of the isolates were recorded. *S.* Dublin was consistently isolated throughout the study period, with the majority (74.4%) of its isolates recorded during the first three 10-year

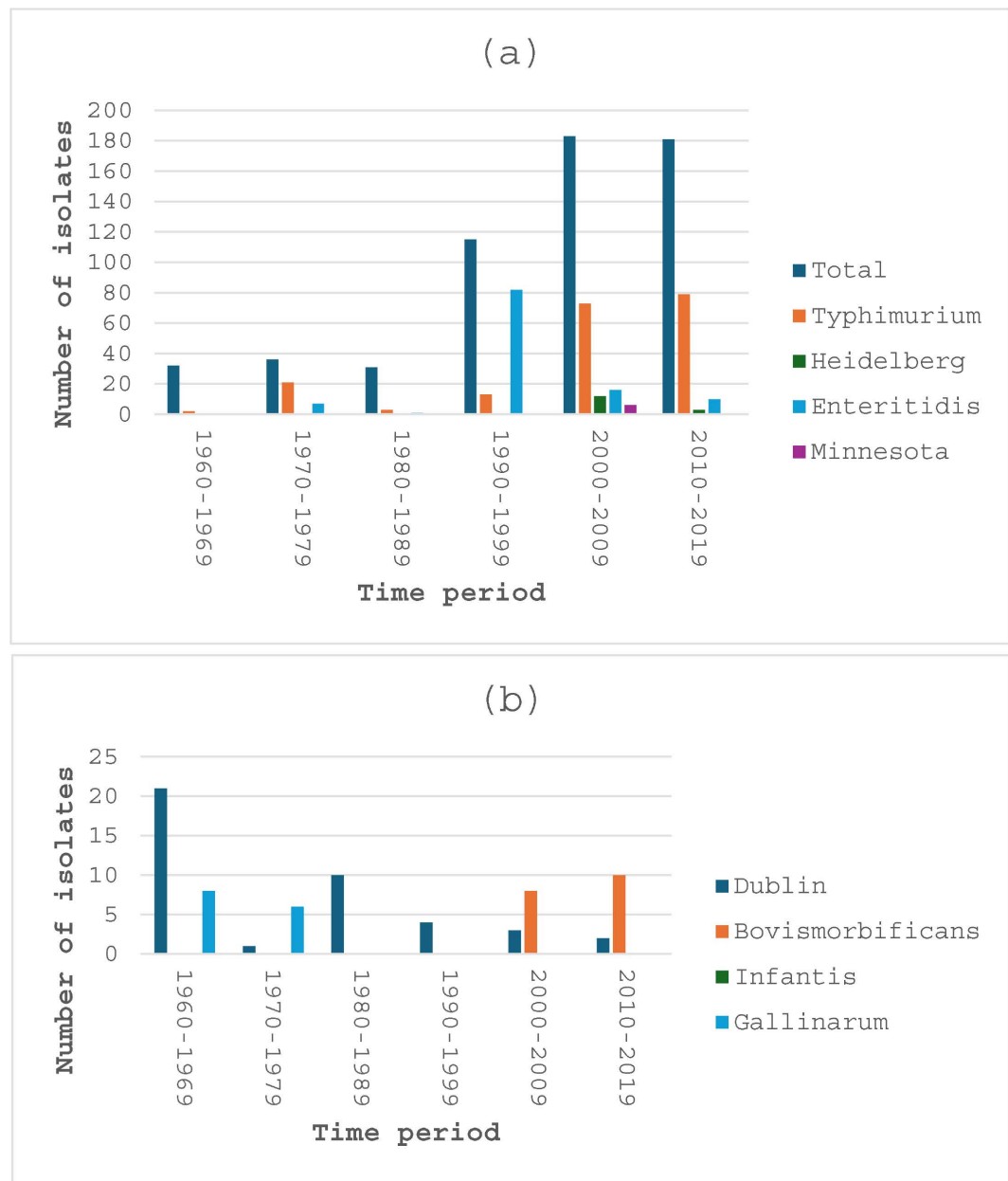

**Fig 1.  Distribution of (a) the total, *S*. Typhimurium, *S*. Heidelberg, *S*. Enteritidis, *S*. Minnesota and (b) *S*. Dublin, *S*. Bovismorbificans, *S*. Infantis and *S*. Gallinarum isolates during the six 10-year study periods.**

periods (Fig 1b). *S*. Bovismorbificans was isolated during the periods of 2000–2009 and 2010–2019, while no isolates of *S*. Infantis were recorded during the six 10-year periods (Fig 1b); all its isolates were detected in the 4-year 2020–2023 period (data not shown). *S*. Gallinarum was isolated during the periods 1960–1969 and 1970–1979 (Fig 1b).

## Overall antibiotic resistance

Table 3 presents the antibiotic resistance prevalence among the *Salmonella* serotypes and isolates tested. Cefoxitin showed the highest percentage of serotypes resistant (65.7%), followed by cephalothin (62.8%) and tetracycline (59.8%).

Aztreonam exhibited the lowest antibiotic resistance among serotypes (11.8%), while sulfamethoxazole-trimethoprim and chloramphenicol recorded the lowest resistance among isolates (9.6%).

Table 4 displays the numbers and percentages of resistant *Salmonella* isolates for eight serotypes with ≥ 10 isolates. *S.* Heidelberg isolates exhibited resistance to all tested antibiotics, ranging from 0.4% for gentamicin to 88.6% for cefoxitin. Resistance surpassed 50% for cefoxitin (88.6%), ceftriaxone (76.3%), tetracycline (75.9%), aztreonam (70.6%), and cephalothin (51.8%). Similarly, *S.* Bovismorbificans isolates demonstrated resistance to all tested antibiotics, with rates ranging from 11.1% for amikacin to 100% for gentamicin. Notably, gentamicin (100%), cephalothin (88.9%), cefoxitin (83.3%), tetracycline (72.2%), and ceftriaxone (55.6%) exhibited over 50% resistance in *S.* Bovismorbificans isolates.

*S.* Enteritidis and *S.* Dublin exhibited resistance to 92.3% (12/13) of the antibiotics tested, with over 50% resistance observed for gentamicin (94.9% and 100%), cefoxitin (89.1% and 76.7%), and cephalothin (59.4% and 58.1%), respectively. *S.* Typhimurium and *S.* Minnesota isolates showed resistance to 84.6% (11/13) and 76.9% (10/13) of the antibiotics tested, respectively. Over 50% of *S.* Typhimurium isolates were resistant to cephalothin (55.6%) and cefoxitin (81.9%), while more than 90% of *S.* Minnesota isolates were resistant to cephalothin, ciprofloxacin, ampicillin, cefoxitin, ceftriaxone, and piperacillin-tazobactam (Table 4). *S.* Infantis isolates were resistant to 69.2% (9/13) of the antibiotics tested, with over 50% resistance observed for four antibiotics: amikacin and piperacillin-tazobactam (68.8%), and cephalothin and cefoxitin (56.3%). *S.* Gallinarum isolates showed resistance to 53.8% (7/13) of the antibiotics tested, with complete resistance (100%) to both gentamicin and cefoxitin. Additionally, more than 80% of *S.* Bovismorbificans isolates were highly resistant to cephalothin, gentamicin, and cefoxitin. Resistance to aztreonam was recorded only in isolates from two serotypes: *S.* Heidelberg (70.6%) and *S.* Bovismorbificans (44.4%), while isolates from the other six serotypes remained fully susceptible to this antibiotic (Table 4).

## Temporal variation in antibiotic resistance during the study period

Fig 2a shows the percentage of *Salmonella* isolates resistant to aztreonam, chloramphenicol, sulfamethoxazole-trimethoprim, ampicillin and amoxicillin-clavulanic acid. Except for a significant (p < 0.01) peak of 14.2% during the period 2000–2009, resistance to aztreonam remained generally very low (<4%) with none recorded during the first two 10-year periods, 1960–1969 and 1970–1979. However, this antibiotic had a significantly (p < 0.001) increased resistance of 35.4% during the 4-year 2020–2023 period (data not shown). The resistance to chloramphenicol was also relatively low (< 13%)

**Table 3. Prevalence of antibiotic resistance among *Salmonella* serotypes (n = 102) and isolates (n = 1028).**

| | | Serotypes | | Isolates | |
|---|---|---|---|---|---|
| **Classes** | **Antibiotic** | **Number resistant** | **% (95% CI)** | **Number resistant** | **% (95% CI)** |
| Cephalosporin | Cefoxitin | 67 | 65.7 (55.6-74.6) | 833 | 81.0 (78.5-83.4) |
| | Cephalothin | 64 | 62.8 (52.6-72.0) | 598 | 58.2 (55.1-61.2) |
| | Ceftriaxone | 41 | 40.2 (30.8-50.4) | 507 | 49.3 (46.2-52.4) |
| Tetracycline | Tetracycline | 61 | 59.8 (49.6-69.3) | 517 | 50.3 (47.2-53.4) |
| Fluoroquinolones | Ciprofloxacin | 54 | 52.9 (42.9-62.8) | 353 | 34.3 (31.5-37.3) |
| Beta-lactamase inhibitors | Amoxicillin-clavulanic acid | 45 | 44.1 (34.4-54.3) | 342 | 33.3 (30.4-36.3) |
| | Piperacillin-tazobactam | 59 | 57.8 (47.6-67.4) | 471 | 45.8 (42.7-48.9) |
| Aminoglycoside | Amikacin | 41 | 40.2 (30.8-50.4) | 179 | 17.4 (15.2-19.9) |
| | Gentamicin | 35 | 34.3 (25.4-44.5) | 273 | 26.6 (23.9-29.4) |
| β-lactams | Aztreonam | 12 | 11.8 (6.5-20.0) | 194 | 18.9 (16.6-21.4) |
| | Ampicillin | 30 | 29.4 (21.0-39.4) | 254 | 24.7 (22.1-27.5) |
| Sulphonamide | Sulfamethoxazole-trimethoprim | 30 | 29.4 (21.0-39.4) | 99 | 9.6 (7.9-11.6) |
| Phenicols | Chloramphenicol | 17 | 16.7 (10.3-25.6) | 99 | 9.6 (7.9-11.6) |

**Table 4. Distribution of resistant isolates for serotypes (n = 8) with ≥ 10 isolates.**

| | Serotype | | | | | | | | | |
|---|---|---|---|---|---|---|---|---|---|---|
| | *S*. Heidelberg (n = 228) | | *S*. Typhimurium (n = 216) | | *S*. Enteritidis (n = 138) | | *S*. Minnesota (n = 108) | | S. Dublin (n = 43) | |
| *Antibiotic | No. resistant | % | No. resistant | % | No. resistant | % | No. resistant | % | No. resistant | % |
| Aztreonam | 161 | 70.6 | 0 | 0.0 | 0 | 0.0 | 0 | 0.0 | 0 | 0.0 |
| Tetracycline | 173 | 75.9 | 62 | 28.7 | 41 | 29.7 | 106 | 98.1 | 8 | 18.6 |
| Cephalothin | 118 | 51.8 | 120 | 55.6 | 82 | 59.4 | 108 | 100.0 | 25 | 58.1 |
| Chloramphenicol | 38 | 16.7 | 16 | 7.4 | 10 | 7.2 | 0 | 0.0 | 10 | 23.3 |
| Ciprofloxacin | 31 | 13.6 | 30 | 13.9 | 51 | 37.0 | 106 | 98.1 | 19 | 44.2 |
| STX | 9 | 3.9 | 21 | 9.7 | 17 | 12.3 | 10 | 9.3 | 2 | 4.7 |
| Ampicillin | 32 | 14.0 | 41 | 19.0 | 14 | 10.1 | 108 | 100.0 | 8 | 18.6 |
| Gentamicin | 1 | 0.4 | 0 | 0.0 | 131 | 94.9 | 0 | 0.0 | 43 | 100.0 |
| Cefoxitin | 202 | 88.6 | 177 | 81.9 | 123 | 89.1 | 108 | 100.0 | 33 | 76.7 |
| Ceftriaxone | 174 | 76.3 | 68 | 31.5 | 38 | 27.5 | 108 | 100.0 | 13 | 30.2 |
| Amikacin | 18 | 7.9 | 23 | 10.6 | 30 | 21.7 | 9 | 8.3 | 17 | 39.5 |
| TZB | 56 | 24.6 | 55 | 25.5 | 61 | 44.2 | 108 | 100.0 | 30 | 69.8 |
| AMC | 85 | 37.3 | 44 | 20.4 | 21 | 15.2 | 100 | 92.6 | 11 | 25.6 |

| | Serotype | | | | | |
|---|---|---|---|---|---|---|
| | *S*. Bovismorbificans (n = 18) | | *S*. Infantis (n = 16) | | *S*. Gallinarium (n = 14) | |
| *Antibiotic | No. resistant | % | No. resistant | % | No. resistant | % |
| Aztreonam | 8 | 44.4 | 0 | 0.0 | 0 | 0.0 |
| Tetracycline | 13 | 72.2 | 0 | 0.0 | 3 | 21.4 |
| Cephalothin | 16 | 88.9 | 9 | 56.3 | 7 | 50.0 |
| Chloramphenicol | 4 | 22.2 | 0 | 0.0 | 0 | 0.0 |
| Ciprofloxacin | 4 | 22.2 | 2 | 12.5 | 5 | 35.7 |
| STX | 3 | 16.7 | 1 | 6.3 | 2 | 14.3 |
| Ampicillin | 3 | 16.7 | 1 | 6.3 | 0 | 0.0 |
| Gentamicin | 18 | 100.0 | 0 | 0.0 | 14 | 100.0 |
| Cefoxitin | 15 | 83.3 | 9 | 56.3 | 14 | 100.0 |
| Ceftriaxone | 10 | 55.6 | 5 | 31.3 | 0 | 0.0 |
| Amikacin | 2 | 11.1 | 11 | 68.8 | 0 | 0.0 |
| TZB | 5 | 27.8 | 11 | 68.8 | 0 | 0.0 |
| AMC | 6 | 33.3 | 4 | 25.0 | 1 | 7.1 |

* STX = Sulfamethoxazole-trimethoprim; TZB = Piperacillin-tazobactam; AMC = Amoxicillin-clavulanic acid.

during the study periods and no resistance was recorded for this antibiotic during the first period (1960–1969). Similarly, the resistance to sulfamethoxazole-trimethoprim remained relatively low (<17%) throughout the study periods with a peak of 16.1% during the third period (1980–1989). Resistance to ampicillin and amoxicillin-clavulanic acid remained below 25% and 30%%, respectively, throughout the six 10-year study periods. Both antibiotics showed a non-significant ($p > 0.05$) decrease in resistance between the first (1960–1969) and second (1970–1979) periods and both of them had significantly ($p < 0.001$) increased resistance (ampicillin, 34.5% and amoxicillin-clavulanic acid, 45.9%) during the 4-year 2020–2023 period (data not shown).

The percentage of *Salmonella* isolates resistant to tetracycline, piperacillin-tazobactam, amikacin and ciprofloxacin is shown in Fig 2b. Resistance to tetracycline significantly ($p < 0.001$) increased above 50% (69.7%) during the 4-year

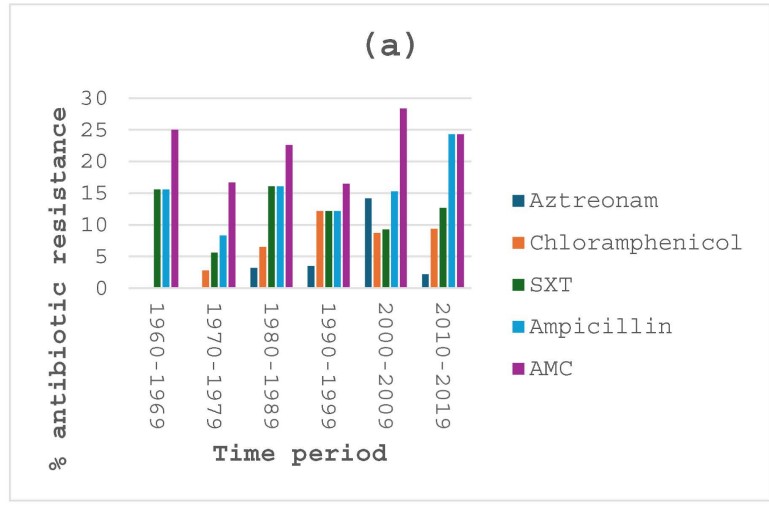

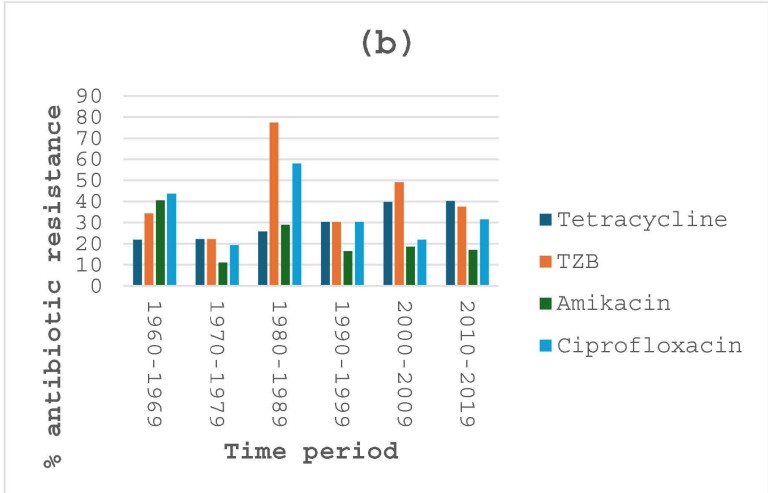

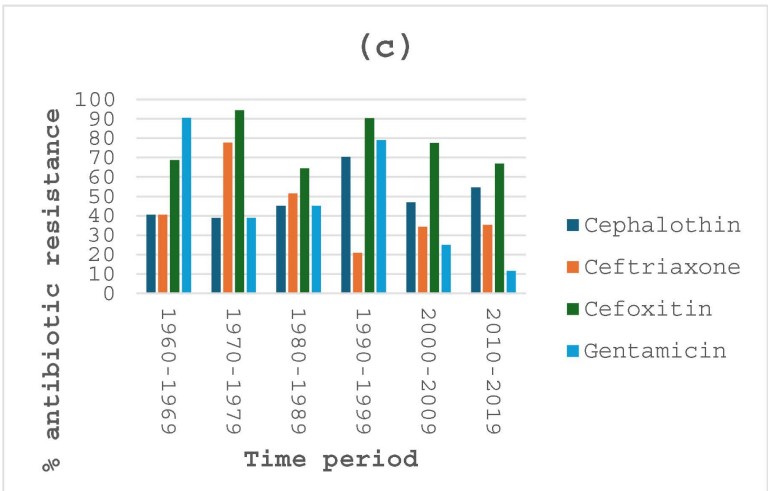

**Fig 2. Percentage antibiotic resistance for (a) aztreonam, chloramphenicol, sulfamethoxazole-trimethoprim (SXT), ampicillin and amoxicillin-clavulanic acid (ACM), (b) tetracycline, piperacillin-tazobactam (TZB), amikacin and ciprofloxacin and (c) cephalothin, ceftriaxone, cefoxitin and gentamicin over the six 10-year periods.**

2020–2023 period (data not shown). Resistance to amikacin significantly decreased (p = 0.01) from 40.6% to 11.1% between the first (1960–1969) and second (1970–1979) periods, increased non-significantly (p > 0.05) to 29% during the third period (1980–1989) and thereafter remaining relatively low between 15.6% and 18.6% for the last three 10-year periods.

## Prevalence of antibiotic resistance genes detected from *Salmonella* serotypes

*S.* Gallinarium recorded, the highest prevalence of antibiotic resistance genes, exceeding 50%, including *blaCMY*-2 (50%), *qnrB* (50%), $bla_{PSE}$ (57.1%), *cat1* (64.3%), *qnrA* (64.3%), and *tetA* (71.4%). The prevalence of antibiotic resistance genes in other serotypes was generally low. The *cm1A* had the highest prevalence for *S.* Typhimurium (32.9%) and *S.* Minnesota (34.3%) isolates, while *sul1* was predominant for *S.* Dublin (37.2%) and *S.* Heidelberg (30.3%) isolates. For *S.* Bovismorbificans, the most prevalent antibiotic resistance gene was *dfrI* (50%), *qrnA* (41.3%) for *S.* Enteritidis, and *cm1A* and *tetA* (18.8%) for *S.* Infantis isolates (Table 5).

## Prevalence of virulence genes identified from *Salmonella* serotypes

Overall, across all serotypes, virulence genes exhibited a prevalence of at least 50%, except for the *srgA* and *gtgE* genes. Virulence genes such as *sspH1, sspH2, sopB, sifA,* and *invA* had a prevalence of ≥ 50%, except among *S.* Minnesota isolates. Similarly, the virulence gene *mig5* was observed to have a prevalence of ≥ 50%, except among *S.* Minnesota and *S.* Typhimurium isolates. Among *S.* Gallinarium isolates, *srgA* and *gtgE* were the only genes absent, while *gtgE* was the only gene with a prevalence of < 50% among Group OMD isolates. Conversely, the virulence genes *spvC, gtgB, gipA, sopE,* and *sodC1* exhibited a prevalence of > 50% only among isolates of these two serotypes. However, no virulence genes were identified in *S.* Minnesota (Table 6).

## The association among antibiotic resistance, antibiotic resistance genes, and the sample source

For all antibiotics tested, there were no significant (p > 0.05) differences in antibiotic resistance between cattle and poultry samples for the *S.* Dublin isolates. The majority of samples for the *S.* Minnesota isolates were from poultry (92.6%, 100/108), with 4 from cattle and 2 from pigs, thus precluding comparisons. Significant differences (p < 0.05) in antibiotic resistance were noted for ceftriaxone and ciprofloxacin for *S.* Typhimurium isolates. Resistance to ceftriaxone was significantly (p = 0.04) higher for poultry (38.7%, 29/75) compared to pig (18.6%, 8/43) isolates, while resistance to ciprofloxacin was also significantly (p = 0.019) higher for pig (20.9%, 9/43) isolates compared to equine (7.1%, 1/14) isolates. (S3 Table).

Three antibiotic resistance genes, $bla_{PSE}$, $bla_{SHV}$, and *flo*, were significantly associated with ceftriaxone resistance in S. Dublin isolates. Resistance to cephalothin was significantly associated with two (*sul1* and *tetA*), three ($bla_{PSE}$, $bla_{SHV}$, and *dfrXIII*), and six ($bla_{CMY-2}$, $bla_{PSE}$, $bla_{SHV}$, *qnrS, sul1,* and *tetA*) resistance genes for *S.* Enteritidis, *S.* Typhimurium, and *S.* Heidelberg isolates, respectively. The antibiotic resistance genes $bla_{PSE}$ and *qnrS* were significantly associated with chloramphenicol resistance in *S.* Enteritidis and *S.* Dublin isolates, respectively. Ciprofloxacin resistance was associated with *flo, cmlA,* and *qnrB* resistance genes in *S.* Heidelberg isolates and none in isolates of other serotypes. None of the resistance genes were associated with gentamicin resistance in *S.* Enteritidis isolates, and no analysis was conducted for isolates of other serotypes, as they were either all susceptible or all resistant to this drug. A significant association was found between Flo and *dfrI* resistance genes and resistance to piperacillin-tazobactam in *S.* Heidelberg isolates. Resistance to sulfamethoxazole-trimethoprim was significantly associated with $bla_{TEM}$, *qnrS,* and *tetA* resistance genes in *S.* Typhimurium isolates; *qnrA* and *dfrXIII* in *S.* Heidelberg isolates; and *qnrB* and *tetA* in *S.* Enteritidis isolates. Tetracycline resistance in *S.* Typhimurium isolates was significantly associated with nine resistance genes ($bla_{CMY-2}$, $bla_{SHV}$, *cat1, qnrA,*

**Table 5. Prevalence (%) of antibiotic resistance genes from *Salmonella* serotypes.**

| Gene | All sero-types (n=102) | All isolates (n=1028) | Typh-imurium (n=216) | Minne-sota (n=108) | Heidel-berg (n=228) | Galli-narium (n=14) | Enter-itidis (n=138) | Dublin (n=43) | Bovis-mor (n=18) | Group OMD (n=16) | Infan-tis (n=16) |
|---|---|---|---|---|---|---|---|---|---|---|---|
| blaTEM | 24.5 | 7.0 | 6.5 | 0.0 | 6.1 | 21.4 | 8.0 | 7.0 | 11.1 | 18.8 | 6.3 |
| blacmy-2 | 44.1 | 21.4 | 12.0 | 31.5 | 20.6 | 50.0 | 18.8 | 20.9 | 22.2 | 81.3 | 12.5 |
| blaPSE | 35.3 | 15.4 | 10.6 | 1.9 | 20.2 | 57.1 | 13.0 | 18.6 | 27.8 | 62.3 | 12.5 |
| bBlashv | 36.3 | 18.1 | 14.8 | 21.3 | 18.0 | 35.7 | 18.1 | 16.3 | 11.1 | 37.5 | 12.5 |
| cat1 | 30.4 | 15.4 | 16.2 | 1.9 | 19.7 | 64.3 | 17.4 | 14.0 | 16.7 | 0.0 | 0.0 |
| Flo | 37.3 | 15.0 | 14.4 | 10.2 | 9.2 | 7.1 | 16.7 | 27.9 | 22.2 | 56.3 | 12.5 |
| cmlA | 54.9 | 26.3 | 32.9 | 34.3 | 13.6 | 21.4 | 20.3 | 25.6 | 38.9 | 56.3 | 18.8 |
| qnrA | 45.1 | 26.1 | 24.5 | 2.8 | 27.6 | 64.3 | 41.3 | 23.3 | 38.9 | 18.8 | 0.0 |
| qnrB | 39.2 | 18.4 | 13.9 | 20.4 | 14.9 | 50.0 | 23.9 | 23.3 | 22.2 | 31.3 | 12.5 |
| qnrS | 29.4 | 15.1 | 16.7 | 5.6 | 16.2 | 21.4 | 20.3 | 20.9 | 11.1 | 12.5 | 0.0 |
| sul1 | 41.2 | 24.5 | 25.0 | 0.0 | 30.3 | 42.9 | 30.4 | 37.2 | 22.2 | 12.5 | 12.5 |
| sul2 | 12.7 | 2.5 | 1.4 | 0.0 | 0.9 | 7.1 | 6.5 | 4.7 | 0.0 | 12.5 | 0.0 |
| sul3 | 13.7 | 6.5 | 6.9 | 0.0 | 4.4 | 7.1 | 13.8 | 23.3 | 16.7 | 0.0 | 0.0 |
| tetA | 38.2 | 23.4 | 22.7 | 0.0 | 29.4 | 71.4 | 29.7 | 27.9 | 38.9 | 25.0 | 18.8 |
| tetB | 33.3 | 16.0 | 19.0 | 0.0 | 11.8 | 21.4 | 25.4 | 16.3 | 16.7 | 31.3 | 6.3 |
| dfrI | 32.4 | 15.8 | 15.7 | 3.7 | 13.6 | 21.4 | 21.7 | 18.6 | 50.0 | 37.5 | 12.5 |
| dfrXII | 26.5 | 8.8 | 6.9 | 0.0 | 7.9 | 21.4 | 8.7 | 9.3 | 33.3 | 37.5 | 12.5 |
| dfrXIII | 21.6 | 9.3 | 10.2 | 1.9 | 6.6 | 21.4 | 16.7 | 9.3 | 22.2 | 0.0 | 0.0 |

**Table 6. Prevalence (%) of virulence genes from *Salmonella* serotypes.**

| Gene | All serotypes (n=102) | All isolates (n=1028) | Typhimurium (n=216) | Minnesota (n=108) | Heidelberg (n=228) | Gallinar-ium (n=14) | Enteritidis (n=138) | Dublin (n=43) | Bovismor (n=18) | Infantis (n=16) |
|---|---|---|---|---|---|---|---|---|---|---|
| sspH1 | 74.5 | 64.4 | 65.7 | 20.4 | 78.5 | 100.0 | 81.9 | 74.4 | 72.2 | 25.0 |
| srgA | 37.3 | 12.1 | 9.3 | 6.5 | 6.1 | 0.0 | 8.7 | 20.9 | 0.0 | 12.5 |
| sspH2 | 77.5 | 62.9 | 76.4 | 30.6 | 62.7 | 64.3 | 60.1 | 81.4 | 55.6 | 75.0 |
| sopB | 85.3 | 69.0 | 78.7 | 22.2 | 66.2 | 85.7 | 79.0 | 83.7 | 77.8 | 100.0 |
| sifA | 75.5 | 63.4 | 64.4 | 16.7 | 72.8 | 92.9 | 81.9 | 74.4 | 72.2 | 50.0 |
| pefA | 66.7 | 43.7 | 46.3 | 3.7 | 48.7 | 57.1 | 46.4 | 58.1 | 50.0 | 50.0 |
| rcK | 63.7 | 36.4 | 29.6 | 17.6 | 46.9 | 57.1 | 27.5 | 48.8 | 11.1 | 37.5 |
| mig5 | 76.5 | 53.5 | 48.6 | 18.5 | 61.8 | 78.6 | 62.3 | 67.4 | 27.8 | 50.0 |
| invA | 100.0. | 100.0 | 100.0 | 100.0 | 100.0 | 100.0 | 100.0 | 100.0 | 100.0 | 100.0 |
| spvC | 61.8 | 36.6 | 38.9 | 10.2 | 37.7 | 71.4 | 34.8 | 46.5 | 38.9 | 50.0 |
| gtgB | 50.0 | 24.0 | 23.6 | 2.8 | 21.5 | 50.0 | 29.0 | 37.2 | 16.7 | 18.8 |
| gipA | 52.0 | 39.1 | 45.4 | 5.6 | 43.0 | 64.3 | 47.1 | 39.5 | 55.6 | 25.0 |
| sopE | 57.8 | 34.4 | 38.0 | 14.8 | 35.1 | 71.4 | 33.3 | 34.9 | 44.4 | 37.5 |
| sodC1 | 64.7 | 38.4 | 47.7 | 20.4 | 30.7 | 64.3 | 35.5 | 51.2 | 27.8 | 43.8 |
| gtgE | 27.5 | 8.9 | 8.3 | 10.2 | 3.9 | 0.0 | 8.7 | 7.0 | 5.6 | 0.0 |

qnrB, qnrS, tetA, dfrXI, and dfrXII), while it was associated with five genes (bla$_{SHV}$, qnrB, qnrS, sul1, and dfrXII) in S. Enteritidis isolates.

## Discussion

The diversity of *Salmonella* strains evaluated in this study is consistent with previous studies reported by [15], and [16] who reported that the most common *Salmonella* serotypes in SA fluctuate over time based on geographical distribution and their capacity to affect different hosts. Despite being isolated only between 2000 and 2009, *S.* Heidelberg emerged as the most prevalent serotype in this study. This observation indicates a recent potential shift in the epidemiology of *Salmonella* serotypes in SA. The elevated prevalence of *S.* Heidelberg in this study is in part attributable to its isolation from imported meat intended for local consumption, highlighting improved efforts in *Salmonella* surveillance from SA port of entry. [20] provided the evidence that *S.* Heidelberg is the most prevalent serotype (73.9%) contaminating mechanically recovered poultry meat imported into SA between 2016 and 2017 which align with the findings our study.

The prevalence of *Salmonella* Typhimurium, and *S.* Enteritidis in the current study aligns with global trends observed across animals, food, environment, and human populations [21,22]. In a recent study utilizing whole-genome sequencing, it was found that *S.* Typhimurium and *S.* Enteritidis strains from SA shared established lineages with a common ancestor dating back to circa 1800's [7], suggesting that these serotypes are well established in the country. Moreover, several studies have indicated *S.* Typhimurium and *S.* Enteritidis as the most frequently isolated serotypes from food, animals, and wildlife in SA over the last six decades [13,15,23]. Since 2012, *S.* Enteritidis and *S.* Typhimurium have ranked as first and second most prevalent non-typhoidal *S. enterica* serotype in human clinical cases in SA [24].

In the current study, *S.* Minnesota emerged as the fourth most prevalent serotype. This frequency coincides with a high level of detection of *S.* Minnesota in imported meat and meat products into SA (unpublished data from the ARC-OVR Bacteriology laboratories) suggesting that SA imports are originating from countries with poor sanitary practices and a systematic lack of policies and procedures. The detection of *S.* Minnesota in imported meat samples at least highlights the robust *Salmonella* monitoring and surveillance at ports of entry and the diagnostic capacity of veterinary and food laboratories in the country.

Other serotypes in this study that showed notable temporal changes in prevalence include *S.* Dublin, *S.* Bovismorbificans, and *S.* Infantis. *S.* Infantis has been receiving increased global attention due to its rising incidences in many countries and the occurrence of high levels of MDR amongst the isolates [25]. In SA, *S.* Infantis is reported amongst the top four serovars responsible for human infections with poultry products, being highlighted as significant source of the bacterium [26].

The high levels of cephalosporin resistance reported in this study aligns with its consistent reporting in antimicrobial resistance monitoring and surveillance studies involving clinical isolates, as well as isolates from food (including retail meat) and animals globally [27–29].

A significant increase in cephalothin resistance (p < 0.001) was noted in *Salmonella* isolates. This trend is worrisome as, in SA, these antibiotics are exclusively accessible under prescription issued by licensed physicians or veterinarians under the Medicines and Related Substances Control Act (101 of 1965) [30] (https://anyflip.com/enio/wsvm). This antibiotic was first registered in SA in 1969 for use in both animals and humans. Consequently, this resistance trend aligns with global patterns [31]. Moreover, this increase mirrors a similar pattern as the third most used antibiotics in the country were third generation cephalosporins of 16.8% in 2018 and 13.3% in 2020 (https://www.healthdata.org/sites/default/files/files/Projects/GRAM/S__Africa_0.pdf).

The resistance of piperacillin-tazobactam and ceftriaxone reported in this study was surprising since these antibiotics are registered in SA for treatment of human infections. This finding highlighted the complexity of the one health paradigm and global trade taking cognisance that these antibiotics are registered in SA for humans use only. Moreover, this resistance, was exacerbated by high resistance (p < 0.001) of ceftriaxone reported in *Salmonella* isolated from poultry (79.3%)

compared to isolates from other animal species. A potential explanation could be attributed to farming practices and unauthorized usage in animals [32].

In the present study, it was evident that seven *Salmonella* serotypes, MDR patterns, demonstrated resistance to three or more classes of antibiotics. This observation aligns with the global trend of MDR emergence in *Salmonella* serotypes, with reports indicating a rise in resistance to third generation cephalosporins [33,34]. The prevalence of MDR *Salmonella* has been notably escalating in the African continent [35], posing significant challenges not only to the treatment of *Salmonella* but might also change the epidemiology of the disease in the future. This trend underscores the urgent need for effective antimicrobial stewardship and surveillance measures to combat the spread of antibiotic resistance in *Salmonella* populations.

Comparative analyses with similar studies screening antimicrobial resistance genes from food animals revealed lower rates of $bla_{PSE}$ and $bla_{CMY-2}$ in *S*. Gallinarium compared to the findings of this study [36,37]. The notable prevalence of beta-lactam degrading genetic determinants is concerning, particularly given that extended beta-lactams are the primary treatment option for salmonellosis in humans. Furthermore, the high prevalence of *tetA*, *cat1*, and *qnrA* in *S*. Gallinarium suggest frequent use of tetracycline, chloramphenicol, and quinolone antibiotics in poultry farms under study. These findings underscore the potential role of agricultural practices in driving the dissemination of antibiotic resistance genes within bacterial populations, highlighting the need for judicious antibiotic use and enhanced surveillance efforts to mitigate the spread of resistance [36]. Moreover, environmental contamination, could act as a reservoir or vector for bacteria carrying these genes [38]. Thus, environmental contamination, coupled with agricultural practices, plays a key role in the persistence and dissemination of antibiotic resistance genes, making it crucial to consider environmental factors in resistance management and mitigation strategies [39,40].

Significant levels of antimicrobial resistance phenotypes were identified in comparison to the resistance genetic determinants, underscoring the potential risks posed to humans by livestock-associated with antibiotic-resistant *Salmonella*. The moderate level of resistance observed towards ciprofloxacin raises concerns regarding its future efficacy. Moreover, the detection of fluoroquinolone-resistant *Salmonella* species, listed as a high priority by the World Health Organization (https://www.who.int/publications/i/item/9789240093461), is particularly worrisome. The presence of the $bla_{CMY-2}$ gene, which confers resistance to cephalosporins, is also alarming given that this antibiotic is among the latest introduced for medicinal use.

The virulence genes detected in this study exemplify the multifaceted strategies employed by *Salmonella* to invade host cells, evade immune defenses, and establish successful infections. The variety of virulence gene profiles detected among the isolates can be attributed to the diverse origins of the isolates and the temporal span of this culture collection. This diversity underscores the importance of continuous monitoring and characterization of virulence factors across various *Salmonella* serotypes to better understand the epidemiology and pathogenic mechanisms of this significant pathogen.

It is also important to note that prolonged storage, especially under unfavourable conditions, can lead to genetic mutations, loss of plasmids, or other genetic changes that may reduce the bacteria's ability to cause disease. This reduction can occur due to the loss of virulence factors such as toxins, adhesion molecules, and enzymes that contribute to the pathogenicity of the bacteria [41] such changes could potentially explain some of the findings in our study.

The primary limitation of this study was the potential bias in the samples submitted for testing, as not all suspected cases of Nontyphoidal *Salmonella* in the country were included. Additionally, some *Salmonella* isolates at ARC: OVR were not preserved during certain periods, which may have affected the observed prevalence of the dominant serotypes. Furthermore, incomplete entries in laboratory records and sample submission forms prevented us from capturing all necessary data, limiting our ability to draw more precise conclusions from the metadata, particularly in relation to source tracking.

## Conclusion

This study revealed a diverse range of *Salmonella* serotypes, with their prevalence fluctuating over time based on geographical distribution and host susceptibility. Temporal analysis of resistance patterns highlighted a general trend of

gradual increase, particularly notable in the third generation cephalosporins, across the last three 10-year periods (1990–1999, 2000–2009 and 2010–2019). Despite the legislative measures across the world, no single control measure has been documented as sufficiently effective in significantly reducing the level of contamination of NT-*Salmonella* strains to acceptable level. This underscores the pressing need for stakeholders in the animal protein food chain to devise alternative approaches to antimicrobial resistance and prudent use of antimicrobials. Future initiatives will aim to engage and collaborate with human medical diagnosticians and researchers to access and analyse *Salmonella* isolates from the human health sector. By embracing One Health Approach in analysing circulating strains in the human, animal, plant, water and the environmental ecosystem, a more comprehensive framework can be established for monitoring and surveillance purposes for better global health security.

## Supporting information

**S1 Data.**
(XLSX)

## Acknowledgments

The authors would like to acknowledge the Onderstepoort Veterinary Research, for granting them permission to use the data for this study. Allerton veterinary laboratory for donating their *Salmonella* isolates for this study.

## Author contributions

**Conceptualization:** Itumeleng Matle.

**Data curation:** Kudakwashe Magwedere.

**Formal analysis:** Davies Mubika Pfukenyi.

**Funding acquisition:** Itumeleng Matle, Mulunda Mwanza, Kudakwashe Magwedere.

**Methodology:** Davies Mubika Pfukenyi, Nozipho Maphori, Nkagiseng Moatshe, Thabo Nkabinde, Annah Motaung, Tracy Schmidt.

**Writing – original draft:** Itumeleng Matle.

**Writing – review & editing:** Tracy Schmidt, Emmanuel Seakamela, Mulunda Mwanza, Lubanza Ngoma, Mohamed Sirdar, Khanyisile R. Mbatha, Kudakwashe Magwedere.

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
