## [Decision Letter · Decision Letter 0]

Dear Dr. Matle,

Thank you for submitting your manuscript to PLOS ONE. After careful consideration, we feel that it has merit but does not fully meet PLOS ONE’s publication criteria as it currently stands. Therefore, we invite you to submit a revised version of the manuscript that addresses the points raised during the review process.

We look forward to receiving your revised manuscript.

Kind regards,

Mabel Kamweli Aworh, DVM, MPH, PhD. FCVSN

Academic Editor

PLOS ONE

Reviewers' comments:

Reviewer's Responses to Questions

**Comments to the Author**

1. Is the manuscript technically sound, and do the data support the conclusions?

Reviewer #1: Partly

Reviewer #2: Yes

Reviewer #3: Yes

2. Has the statistical analysis been performed appropriately and rigorously?

Reviewer #1: No

Reviewer #2: Yes

Reviewer #3: Yes

3. Have the authors made all data underlying the findings in their manuscript fully available?

Reviewer #1: Yes

Reviewer #2: Yes

Reviewer #3: Yes

4. Is the manuscript presented in an intelligible fashion and written in standard English?

Reviewer #1: No

Reviewer #2: Yes

Reviewer #3: Yes

Reviewer #1: The authors have presented an important analysis looking at the changing distributions of different NTS serovars in South Africa, and the associated AMR. However, I believe that the manuscript needs complete revision in order to really contribute to knowledge in this area.

General comments

1. The manuscript goes into extreme detail regarding the predominant serotypes and AMR and much of this can be combined into a single argument, including the tables of resistance, resistant genes and serotypes, shortening the manuscript, especially as much of the data can be presented in figures/tables.

2. The authors have alluded to the source of the isolates - but have not provided any information on source attribution. Given the importance of this in NTS management, the authors should analyse which serotypes are associated with which food animals and include these data in the manuscript.

3. The authors refer to "Group OMD as the 8th most common serotype. This is incorrect. S. Gallinarum is the 8th most common serotype - the data needs to be reanalyzed with this in mind because OMD represents a collection of different serotypes, that agglutinate in group specific antisera.

4. Figure 1 - 3 are presented separately, but can be collapsed into a single figure reflecting the 8 most common serotypes (excluding OMD); the same applies to figures 4 and 5.

5. Table 3 should be revised to exclude OMD as a serotype.

6. The authors, for convenience I suspect, divided their data into 13-year periods. I find this artificial and it obstructs from comparing these data to similar data that may be published elsewhere. They should rather use 5 - year or at least 10-year periods, and highlight that before 1960 and after 2020, the data are not representative of a full five years/decade.

Specific comments

1. The authors need to completely revise the referencing. The numbers in the text do not correspond with the references which are presented alphabetically.

2. The authors should ensure that they have correctly italicised the genes referred to in the text.

3. Line 76 to 77: BETA-lactamase is not an antibiotic - it is an enzyme that breaks down BETA-lactam antibiotics.

4. Lines 119 - 121: this does not make sense and should be revised.

5. Please include the correct website address at line 158 - 159, instead of this version?

6. The PCR methodology does not need to be described in full - the authors can state "as previously described and provide a reference.

7. Line 473 ssspH1 should be corrected to sspH.

8. Line 546 & 565 - State the name of the author of each publication, not just the (incorrect) reference numbers [27] and [31]. This rule applies throughout.

9. Line 654. Delete the comma after patterns to give this sense.

10. Line 661 - 662. Serotype severity does not differ between hosts, the severity of disease due to different serotypes does.

11. Lines 676 - 681 refer to antimicrobial use in chickens, but what is the potential role of environmental contamination in selecting for resistance?

12. Lines 701 - 708. Some of the isolates analysed go back to the 1950s. Can the authors be confident that their isolates did not lose virulence characteristics over the time period?

Reviewer #2: Dear Authors,

You have done a great job and I learned a lot from your manuscript.

However, I have a few revisions to ask of you.

Line 562 states “… in an earlier study by [27].” Please add the name(s) of the author(s) being cited.

Line 564 says “[31] provided the evidence …” Please edit it as requested above.

Line 588 should start as a new sentence i.e. “On the grounds of these findings, …”.

Lines 631 to 632 are incomplete. “The resistance of piperacillin-tazobactam and ceftriaxone reported in this study was surprising since these antibiotics are registered in SA for human.” Human what?

Well done!

Reviewer #3: The manuscript addresses an important global issue, focusing on the monitoring, surveillance, antimicrobial resistance, and genetic diversity analysis of non-typhoidal Salmonella (NTS) in South Africa from 1959 to 2023, specifically from animals and animal products. A few comments:

Materials and Methods

While the total number of NTS isolates (1,028) is mentioned in the abstract, this detail is missing from the Materials and Methods section. It would be beneficial for the authors to specify how many isolates were obtained from animals, the environment, and food products, respectively. Additionally, it is unclear if there are differences in the types of Salmonella serovars detected across these sources. Including this information in a table would greatly enhance clarity and provide readers with a more comprehensive understanding of the study's findings. Although the association among antibiotic resistance, antibiotic resistance genes, and the sample source was mentioned.

**Do you want your identity to be public for this peer review?** For information about this choice, including consent withdrawal, please see our Privacy Policy

Reviewer #1: No

Reviewer #2: **Yes: ** Dr. Oluwafolayemi Doyeni

Reviewer #3: No

---

## [Author Response · Author response to Decision Letter 1]

8 Mar 2025

Reviewer # 1 comments

The manuscript goes into extreme detail regarding the predominant serotypes and AMR and much of this can be combined into a single argument, including the tables of resistance, resistant genes and serotypes, shortening the manuscript, especially as much of the data can be presented in figures/tables.

Response 1

Thank you for your feedback. While we understand the suggestion to streamline the manuscript, we believe that the detailed discussion of predominant serotypes, antibiotic resistance (AMR), and their associated resistance genes is crucial for providing comprehensive insights into the study's findings. Each section offers unique contributions to the understanding of the complex relationships between Salmonella serotypes and antimicrobial resistance profiles, and combining these aspects may compromise the clarity and depth of the analysis.

Furthermore, while we agree that data can be effectively presented in figures and tables, the narrative explanation of the findings is necessary to provide context, interpretation, and highlight the significance of the results. The current structure aims to balance data presentation with in-depth analysis, ensuring that readers can appreciate both the broader trends and the nuances of the study.

We believe this approach enhances the manuscript’s value, and we are open to any suggestions for further conciseness without sacrificing the integrity of the information presented.

Line 206 The authors have alluded to the source of the isolates - but have not provided any information on source attribution. Given the importance of this in NTS management, the authors should analyse which serotypes are associated with which food animals and include these data in the manuscript.

Kindly refer to line 249 -307. The author provide the link between the major serotypes to the source of isolation.

Line 211-215 The authors refer to "Group OMD as the 8th most common serotype. This is incorrect. S. Gallinarum is the 8th most common serotype - the data needs to be reanalyzed with this in mind because OMD represents a collection of different serotypes, that agglutinate in group specific antisera.

The data has been reanalyzed to reflect S. Gallinarum as the eighth most common serotype, and **Group OMD** has been removed throughout the manuscript wherever prevalence was described or discussed.

Line 220-223 Figure 1 - 3 are presented separately, but can be collapsed into a single figure reflecting the 8 most common serotypes (excluding OMD); the same applies to figures 4 and 5.

Figure 1–3 have been merged into a single figure, now referred to as “Figure 1, titled:

"Distribution of (a) total isolates, S. Typhimurium, S. Heidelberg, S. Enteritidis, and S. Minnesota, and (b) S. Dublin, S. Bovismorbificans, S. Infantis, and S. Gallinarum across six 10-year study periods."

Line 253-260 Table 3 should be revised to exclude OMD as a serotype.

Table 3 has been renamed Table 4 and revised to exclude OMD as a serotype.

The authors, for convenience I suspect, divided their data into 13-year periods. I find this artificial and it obstructs from comparing these data to similar data that may be published elsewhere. They should rather use 5 - year or at least 10-year periods, and highlight that before 1960 and after 2020, the data are not representative of a full five years/decade.

The data has been analyzed in 10-year periods from lines 251 to 254, and all subsequent analyses followed this timeframe. Corrections have been made throughout the manuscript accordingly.

The authors need to completely revise the referencing. The numbers in the text do not correspond with the references which are presented alphabetically. Thank you for the comment. The authors have revised the references accordingly

The authors should ensure that they have correctly italicised the genes referred to in the text The authors have italicised all genes throughout the document

3. Line 76 to 77: Beta-lactamase is not an antibiotic - it is an enzyme that breaks down BETA-lactam antibiotics Thank you and that’s correct. The statement has been deleted.

Lines 119 - 121: this does not make sense and should be revised The sentence has been revised in line 120-122 and reads as follows “In South Africa, the Animal Diseases and Parasites Act (1956) mandates that S. Enteritidis isolated from all fowls, including poultry and ostriches, must be reported to the state authority.

Please include the correct website address at line 158 - 159, instead of this version? Corrected accordingly

The PCR methodology does not need to be described in full - the authors can state "as previously described and provide a reference. Thank you for the comment. The PCR method was corrected as advised.

Line 473 ssspH1 should be corrected to sspH Corrected accordingly

Line 546 & 565 - State the name of the author of each publication, not just the (incorrect) reference numbers [27] and [31]. This rule applies throughout Corrected accordingly

Line 654. Delete the comma after patterns to give this sense Comma deleted as advised

Line 661 - 662. Serotype severity does not differ between hosts, the severity of disease due to different serotypes does The statement has been deleted

Lines 676 - 681 refer to antimicrobial use in chickens, but what is the potential role of environmental contamination in selecting for resistance? Kindly refer to line 679 -689 where the authors added the contribution of environmental contamination.

Lines 701 - 708. Some of the isolates analysed go back to the 1950s. Can the authors be confident that their isolates did not lose virulence characteristics o Its correct that bacteria can loss virulence characteristics over time particular if there are stored under unfavarable conditions for the long time. The authors acknowledge this and have responded to it in line 710-714

Reviewer # 2 comments

Response

Line 562 states “… in an earlier study by [27].” Please add the name(s) of the author(s) being cited. Corrected accordingly

Line 564 says “[31] provided the evidence …” Please edit it as requested above.

Corrected accordingly

Line 588 should start as a new sentence i.e. “On the grounds of these findings, …”.

Corrected as advised

Lines 631 to 632 are incomplete. “The resistance of piperacillin-tazobactam and ceftriaxone reported in this study was surprising since these antibiotics are registered in SA for human.” Human what? Corrected accordingly

Reviewer # 2 comments

Response

Line 248-251 While the total number of NTS isolates (1,028) is mentioned in the abstract, this detail is missing from the Materials and Methods section. It would be beneficial for the authors to specify how many isolates were obtained from animals, the environment, and food products, respectively. Additionally, it is unclear if there are differences in the types of Salmonella serovars detected across these sources. Including this information in a table would greatly enhance clarity and provide readers with a more comprehensive understanding of the study's findings. Although the association among antibiotic resistance, antibiotic resistance genes, and the sample source was mentioned.

The information on the source of isolation and its link to the major serotypes has been added as part of **Table 2**.

---

## [Decision Letter · Decision Letter 1]

Dear Dr. Matle,

Thank you for submitting your manuscript to PLOS ONE. After careful consideration, we feel that it has merit but does not fully meet PLOS ONE’s publication criteria as it currently stands. Therefore, we invite you to submit a revised version of the manuscript that addresses the points raised during the review process.

We look forward to receiving your revised manuscript.

Kind regards,

Mabel Kamweli Aworh, DVM, MPH, PhD. FCVSN

Academic Editor

PLOS ONE

**Journal Requirements:**

**Additional Editor Comments:**

In addition to addressing the reviewers' comments, kindly fix the following issues:

1. Please ignore reviewer 2's comment to change "virulence" to "virulent" and also in line 494 "virulence genes" to "virulent genes."

2. The pages of the revised manuscript should be numbered.

3. Kindly reduce the introduction section significantly to a maximum of two pages. 

4. Summarize the text in lines 371-427 with appropriate tables. If this has been done already, kindly delete the text to avoid redundancy, as this manuscript is way too long. The information in lines 429-460 has been represented in figure 2'. Please reduce the text to a few sentences highlighting key results to avoid duplication of information already captured and redundancy.

5. Summarize the text in lines 521-553 with a table and reduce the text significantly, highlighting only important results in a few sentences.

6. Please reduce the discussion section significantly to a maximum of 3 pages discussing only major study results. Move the limitations to the last paragraph of the discussion section. Please delete the subheading "Study limitations".

Reviewers' comments:

Reviewer's Responses to Questions

**Comments to the Author**

Reviewer #1: (No Response)

Reviewer #2: (No Response)

2. Is the manuscript technically sound, and do the data support the conclusions?

Reviewer #1: Yes

Reviewer #2: Yes

3. Has the statistical analysis been performed appropriately and rigorously?

Reviewer #1: Yes

Reviewer #2: Yes

4. Have the authors made all data underlying the findings in their manuscript fully available?

Reviewer #1: Yes

Reviewer #2: Yes

5. Is the manuscript presented in an intelligible fashion and written in standard English?

Reviewer #1: Yes

Reviewer #2: Yes

**Reviewer #1: ** While the authors have addressed all the comments, I believe that they should still add information in the text and in table 2 on source attribution. Rather than say that Salmonella was isolated from meat, they should specify the type of meat (i.e. chicken, pork beef etc) as this has implications for disease control. If these data were not captured they should address this in the text and the limitations.

**Reviewer #2:**  Dear Authors,

Your efforts on this manuscript are acknowledged and commended.

I’ll start by agreeing with Reviewer#1 that this manuscript could have been more compact and easier to follow without losing relevant details if you had used tables and graphs instead of lengthy prose.

What is the rationale behind the years you chose to review, especially 1959? I expected to read that in your methodology but may have missed it.

Line 41 - 87% of the virulent (not virulence) genes screened …

Line 43 -... resistance, driven by misuse (of what), …

Lines 43 to 44 -... could alter serotype dynamics (remove the comma) and change (not changing) disease epidemiology.

Line 54 -... cause cross-infection between animals and humans [3]. (end your sentence here and begin another) 99% of human serotypes belong to …

Line 73 -... along with at least (why “at least”) amoxicillin clavulanic acid ...

Line 77 - … towards fluoroquinolones (remove the comma) and third-generation cephalosporins.

Line 97 -... operational in SA (write in full at first mention with the abbreviation in parentheses) …

Line 103 - delete “virology” as it is already included in “microbiology”.

Line 113 - “… in the 1930s from …” not from the 1930’s.

Line 494 - “virulent genes” not “virulence genes”

I read your response to Reviewer#1. My topmost recommendation is that you should take their advice to improve the quality of your work. Think of presenting this on a global platform. How will you make a great impression on your audience without losing their attention?

Thank you!

**Do you want your identity to be public for this peer review?** For information about this choice, including consent withdrawal, please see our Privacy Policy

Reviewer #1: No

Reviewer #2: **Yes: ** Dr. Oluwafolayemi Doyeni

---

## [Author Response · Author response to Decision Letter 2]

25 May 2025

Journal Requirements:

All the responses in this rebuttal letter, including the corresponding line numbers, refer to the revised manuscript with track changes

Comment 1: Please review your reference list to ensure that it is complete and correct. If you have cited papers that have been retracted, please include the rationale for doing so in the manuscript text, or remove these references and replace them with relevant current references. Any changes to the reference list should be mentioned in the rebuttal letter that accompanies your revised manuscript. If you need to cite a retracted article, indicate the article’s retracted status in the References list and also include a citation and full reference for the retraction notice.

Respond 1: We have reviewed the status of the papers listed in our references, specifically checking for any retraction notices. As of now, none of the references provided are flagged as retracted in major databases such as Retraction Watch, PubMed, or on the publishers’ websites. If the journal has identified any such papers, kindly provide us with the relevant details, and we will be happy to address the matter promptly.

Additional Editor Comments:

In addition to addressing the reviewers' comments, kindly fix the following issues:

Comment 2: Please ignore reviewer 2's comment to change "virulence" to "virulent" and also in line 494 "virulence genes" to "virulent genes."

Respond 2: Duly noted, the authors didn’t change suggested by the editors

Comment 3. The pages of the revised manuscript should be numbered.

Respond 3: Corrected

Comment 4. Kindly reduce the introduction section significantly to a maximum of two pages.

Respond 4: Reduced to the required pages

Comment 5. Summarize the text in lines 371-427 with appropriate tables. If this has been done already, kindly delete the text to avoid redundancy, as this manuscript is way too long. The information in lines 429-460 has been represented in figure 2'. Please reduce the text to a few sentences highlighting key results to avoid duplication of information already captured and redundancy.

Respond 5: Deleted. The text has reduced in line 386 – 399. Please note that redundancy from 404 to 444 has been deleted.

Comment 6: Summarize the text in lines 521-553 with a table and reduce the text significantly, highlighting only important results in a few sentences.

Respond 6: The results are summarized in Table 6, while the findings related to virulence genes are presented only in paragraph form.

Comment 7. Please reduce the discussion section significantly to a maximum of 3 pages discussing only major study results. Move the limitations to the last paragraph of the discussion section. Please delete the subheading "Study limitations".

Respond 7: The subheading has been removed from line 797. Additionally, the discussion section has been substantially shortened to three and a half pages, including the paragraph on the study’s limitations.

Comment 8: Reviewer #1: While the authors have addressed all the comments, I believe that they should still add information in the text and in table 2 on source attribution. Rather than say that Salmonella was isolated from meat, they should specify the type of meat (i.e. chicken, pork beef etc) as this has implications for disease control. If these data were not captured, they should address this in the text and the limitations.

Respond 9: Incomplete entries in laboratory records and sample submission forms prevented us from capturing all necessary data, limiting our ability to draw more precise conclusions from the metadata, particularly in relation to source tracking. This point has been addressed in line 777-778

Reviewer #2: Dear Authors,

Your efforts on this manuscript are acknowledged and commended. I’ll start by agreeing with Reviewer#1 that this manuscript could have been more compact and easier to follow without losing relevant details if you had used tables and graphs instead of lengthy prose.

Comment 10: What is the rationale behind the years you chose to review, especially 1959? I expected to read that in your methodology but may have missed it.

Respond 11: As there were only a few isolates from 1950, the author has removed them from the analysis.

Comment 12: Line 41 - 87% of the virulent (not virulence) genes screened …

Respond 12: Please ignore reviewer 2's comment to change "virulence" to "virulent" and also in line 494 "virulence genes" to "virulent genes."

Comment 13: Line 43 -... resistance, driven by misuse (of what), …

Respond 13: Corrected in line 44

Comment 14: Lines 43 to 44 -... could alter serotype dynamics (remove the comma) and change (not changing) disease epidemiology

Respond 14: Corrected in line 45

Comment 15: Line 54 -... cause cross-infection between animals and humans [3]. (end your sentence here and begin another) 99% of human serotypes belong to …

Respond 15: Corrected in line 55

Comment 16: Line 73 -... along with at least (why “at least”) amoxicillin clavulanic acid ...

Respond 16: The sentence has been removed as part of the manuscript reduction.

Comment 17: Line 77 - … towards fluoroquinolones (remove the comma) and third-generation cephalosporins.

Respond 17: The sentence has been removed as part of the manuscript reduction.

Comment 18: Line 97 -... operational in SA (write in full at first mention with the abbreviation in parentheses) …

Respond 18: Corrected in line111

Comment 19: Line 103 - delete “virology” as it is already included in “microbiology”.

Respond 19: The sentence has been removed as part of the manuscript reduction.

Comment 20: Line 113 - “… in the 1930s from …” not from the 1930’s.

Respond 20: Corrected 129 in line

Comment 21: Line 494 - “virulent genes” not “virulence genes”

Thank you!

---

## [Decision Letter · Decision Letter 2]

Dear Dr. Matle,

Thank you for submitting your manuscript to PLOS ONE. After careful consideration, we feel that it has merit but does not fully meet PLOS ONE’s publication criteria as it currently stands. Therefore, we invite you to submit a revised version of the manuscript that addresses the points raised during the review process.

We look forward to receiving your revised manuscript.

Kind regards,

Mabel Kamweli Aworh, DVM, MPH, PhD. FCVSN

Academic Editor

PLOS ONE

**Journal Requirements:**

**Additional Editor Comments:**

In addition to addressing the reviewers' comments, please move the **limitations** to the **last paragraph of the Discussion** and **remove the subtitle "Study Limitations."**

Reviewers' comments:

Reviewer's Responses to Questions

**Comments to the Author**

Reviewer #1: All comments have been addressed

Reviewer #2: All comments have been addressed

2. Is the manuscript technically sound, and do the data support the conclusions?

Reviewer #1: Yes

Reviewer #2: (No Response)

3. Has the statistical analysis been performed appropriately and rigorously?

Reviewer #1: N/A

Reviewer #2: (No Response)

4. Have the authors made all data underlying the findings in their manuscript fully available?

Reviewer #1: Yes

Reviewer #2: (No Response)

5. Is the manuscript presented in an intelligible fashion and written in standard English?

Reviewer #1: Yes

Reviewer #2: (No Response)

**Reviewer #1:**  I have one small comment - DALRRD is now Department of Agriculture. While I accept the work was probably done before the name change, I think it is worthwhile mentioning this under the financial support, for readers who are unaware of this. The manuscript need not go out to review again but this small addition can be made to the editor's satisfaction.

**Reviewer #2:**  (No Response)

**Do you want your identity to be public for this peer review?** For information about this choice, including consent withdrawal, please see our Privacy Policy

Reviewer #1: No

Reviewer #2: **Yes: ** Dr. Oluwafolayemi Doyeni

---

## [Author Response · Author response to Decision Letter 3]

20 Jun 2025

A rebuttal letter

Comment 1: Journal Requirements: Please review your reference list to ensure that it is complete and correct. If you have cited papers that have been retracted, please include the rationale for doing so in the manuscript text or remove these references and replace them with relevant current references. Any changes to the reference list should be mentioned in the rebuttal letter that accompanies your revised manuscript. If you need to cite a retracted article, indicate the article’s retracted status in the References list and also include a citation and full reference for the retraction notice.

Responds 1: Thank you for the comment. The references have been revised, corrected, and completed both in-text and in the bibliography. No retracted papers were cited in this study. Due to the reduction in the manuscript length, some references were removed, which reduced the overall number of citations.

Comment 2: Additional Editor Comments: In addition to addressing the reviewers' comments, please move the limitations to the last paragraph of the Discussion and remove the subtitle "Study Limitations."

Responds 2: The necessary correction has been implemented on page 27, line 569 of the revised manuscript.

Comment 3: Reviewer #1: I have one small comment - DALRRD is now Department of Agriculture. While I accept the work was probably done before the name change, I think it is worthwhile mentioning this under the financial support, for readers who are unaware of this. The manuscript need not go out to review again but this small addition can be made to the editor's satisfaction.

Responds 3: The necessary correction has been implemented on page 27, line 613 of the revised manuscript.

Comment 4: If applicable, we recommend that you deposit your laboratory protocols in protocols.io to enhance the reproducibility of your results. Protocols.io assigns your protocol its own identifier (DOI) so that it can be cited independently in the future. For instructions see: https://journals.plos.org/plosone/s/submission-guidelines#loc-laboratory-protocols. Additionally, PLOS ONE offers an option for publishing peer-reviewed Lab Protocol articles, which describe protocols hosted on protocols.io. Read more information on sharing protocols at https://plos.org/protocols?utm_medium=editorial-email&utm_source=authorletters&utm_campaign=protocols.

Respond 4: Your recommendations are duly noted. However, our laboratory protocols are considered intellectual property of the organization and are reserved for commercial purposes.

---

## [Decision Letter · Decision Letter 3]

Monitoring, surveillance, antimicrobial resistance and genetic diversity analysis of non-typhoidal Salmonella in South Africa from 1959-2023 from animal and animal products.

PONE-D-24-42273R3

Dear Dr. Matle,

We’re pleased to inform you that your manuscript has been judged scientifically suitable for publication and will be formally accepted for publication once it meets all outstanding technical requirements.

Kind regards,

Mabel Kamweli Aworh, DVM, MPH, PhD. FCVSN

Academic Editor

PLOS ONE

Additional Editor Comments (optional):

Reviewers' comments:

Reviewer's Responses to Questions

**Comments to the Author**

Reviewer #1: All comments have been addressed

Reviewer #2: All comments have been addressed

2. Is the manuscript technically sound, and do the data support the conclusions?

Reviewer #1: Yes

Reviewer #2: (No Response)

3. Has the statistical analysis been performed appropriately and rigorously?

Reviewer #1: I Don't Know

Reviewer #2: (No Response)

4. Have the authors made all data underlying the findings in their manuscript fully available?

Reviewer #1: Yes

Reviewer #2: (No Response)

5. Is the manuscript presented in an intelligible fashion and written in standard English?

Reviewer #1: Yes

Reviewer #2: (No Response)

Reviewer #1: I have no further comments or suggestions. The authors have met all the reviewers' recommendations.

Reviewer #2: (No Response)

**Do you want your identity to be public for this peer review?** For information about this choice, including consent withdrawal, please see our Privacy Policy

Reviewer #1: No

Reviewer #2: **Yes: ** Dr. Oluwafolayemi Doyeni
